# Optimising Bioprinting Nozzles through Computational Modelling and Design of Experiments

**DOI:** 10.3390/biomimetics9080460

**Published:** 2024-07-29

**Authors:** Juan C. Gómez Blanco, Antonio Macías-García, Jesús M. Rodríguez-Rego, Laura Mendoza-Cerezo, Francisco M. Sánchez-Margallo, Alfonso C. Marcos-Romero, José B. Pagador-Carrasco

**Affiliations:** 1Jesús Usón Minimally Invasive Surgery Centre, Carretera N-521, km41.8, 10071 Cáceres, Spain; 2Department of Mechanical, Energy and Materials Engineering, School of Industrial Engineering, University of Extremadura, Avenida de Elvas, s/n, 06006 Badajoz, Spainlmencer@unex.es (L.M.-C.); acmarcos@unex.es (A.C.M.-R.)

**Keywords:** bioprinting, microextrusion, computational simulation, nozzle

## Abstract

3D bioprinting is a promising technique for creating artificial tissues and organs. One of the main challenges of bioprinting is cell damage, due to high pressures and tensions. During the biofabrication process, extrusion bioprinting usually results in low cell viability, typically ranging from 40% to 80%, although better printing performance with higher cell viability can be achieved by optimising the experimental design and operating conditions, with nozzle geometry being a key factor. This article presents a review of studies that have used computational fluid dynamics (CFD) to optimise nozzle geometry. They show that the optimal ranges for diameter and length are 0.2 mm to 1 mm and 8 mm to 10 mm, respectively. In addition, it is recommended that the nozzle should have an internal angle of 20 to 30 degrees, an internal coating of ethylenediaminetetraacetic acid (EDTA), and a shear stress of less than 10 kPa. In addition, a design of experiments technique to obtain an optimal 3D bioprinting configuration for a bioink is also presented. This experimental design would identify bioprinting conditions that minimise cell damage and improve the viability of the printed cells.

## 1. Introduction

3D printing is an additive manufacturing technology that makes it possible to create physical objects from a 3D model. In bioprinting, this technology is used to create artificial tissues and organs.

However, there are still many obstacles to ensure cell survival and good printability in extrusion-based 3D bioprinting, where cell viability is among the lowest due, in part, to shear stress in the printing nozzle [1].

In recent years, several articles have discussed the optimisation of extrusion bioprinting using computational simulations [1,2,3,4,5].

Computational fluid dynamics (CFD) is a simulation technique that allows the behaviour of bioinks to be studied by computer at different stages of the bioprinting process [3]. CFD has been used since the late 1950s by computer software for 3D transient applications and for incompressible flow [6], and includes pre-processing, resolution, and post-processing steps, which are used consistently across different programs and additional methods [7].

This system, combined with various discretisation methods, is used by different programs to analyse multiple aspects of various printing methods. The simulation of material flow in bioprinting processes facilitates the understanding of the interrelationships between the required geometry, printing parameters, material properties, and mechanical forces during extrusion, allowing for the prediction of the mechanical stress exerted on the cells and the shape of the extruded filament, as well as helping to optimise the printing process and develop new bioink compositions, thereby reducing the need for extensive experimental work [3].

Thus, CFD has numerous applications in the field of bioprinting, including the following [7]:The printability of bioink has been studied in different bioprinting methods—extrusion, droplet, and laser. In extrusion printing, the ability of the bioink to extrude, and maintain its shape and accuracy is evaluated [8]. For droplet-based printing, the accuracy of the generated droplets is considered [9], and in laser printing, the uniformity and accuracy of the jet [10]. CFD can improve printability by analysing factors such as print speed, nozzle geometry, dispensing pressure and rheological characteristics of the bioink.

Cell viability and nozzle design: Cells are constantly surrounded by stimuli that provide them with information about their environment and allow them to make changes in response. For example, in 2D mesenchymal stem cell cultures, physical conditions such as substrate stiffness influence their morphology, adhesion, and differentiation [11], although this response varies according to cell type. Even routine processes such as pipetting and shaking the flask after trypsinisation expose the cells to shear and hydrodynamic stress [12]. They can also locally sense living tissues through mechanical forces and physical microenvironments, and this sensing can induce in response the regulation of cell growth, differentiation, shape changes, or cell death [13]. Thus, when stress increases, cytoplasmic proteins are recruited along with cytoskeletal proteins [14]. In this way, the signal propagates from the cytoskeletal proteins to the nucleus [15], and the cells can sense and respond to changes in their environment. While these seemingly insignificant conditions may produce these changes in the cells, the various mechanical forces to which they are subjected during the bioprinting process play a crucial role in controlling their growth, differentiation, and motility [16]. In extrusion-based bioprinting, shear stress is particularly important, as it is the main cause of cell damage or death [17], so that higher values of shear stress correlate with lower viable cell populations [18]. Moreover, the duration of exposure to shear stress also affects cell viability, with longer residence times at high levels of shear stress leading to more cell damage [19]. Despite the negative effect of the pressures exerted by the bioink on the cells as they pass through the nozzle during the extrusion bioprinting process, mechanical stresses have been shown to play an important role in the control of cell growth, motility, and contraction [16]. For example, constant shear stress under the appropriate conditions regulates the development of stem cells, differentiating them into cells that experience high mechanical forces in vivo, such as endothelial cells and bone-producing cells [20]. This type of mechanical stress is unavoidable in all dispensing bioprinting techniques and is influenced by multiple parameters, each with a different effect on cell biology. The intensity of the shear stress can have varying effects on the cells, directly affecting their viability during and after bioprinting and thus the functionality of the printed structure. However, it is possible to model this stress mathematically and computationally to optimise the printing process and maximise the number of viable cells in the long term. Nozzle geometry significantly affects cell viability due to shear stress, especially in extrusion-based bioprinting [2,21,22]. Mechanical forces are essential in the development and function of various organs and tissues and have been shown to have a substantial effect on stem cell differentiation and functionality. However, the shear stress produced during the printing process has a direct negative impact on cell viability [22]. During the processing stage of bioprinting, cells are exposed to various mechanical forces, and research has shown that these forces play a crucial role in regulating cell growth, differentiation, and motility [16,23,24,25,26]. Shear stress is directly related to nozzle inlet pressure, where higher pressure increases shear stress and affects cells closer to the wall more than those in the centre, exponentially decreasing cell viability [27]. Inlet pressures that are too low or too high result in insufficient or excessive bioink deposition, respectively [22]. The tapered conical nozzle has a lower maximum wall shear stress value compared to the conical nozzle, except for when the dispensing pressure is 0.025 MPa for a given outlet diameter [28]. In contrast, in the cylindrical nozzle, the entire cylindrical portion of the nozzle experiences a higher shear stress in the wall, which is indicative of the reduced cell viability cylindrical nozzles compared to conical nozzles [19,29]. For example, in a study by Nair et al., a 25 G (0.53 mm) needle was found to have a viability of approximately 90% [30]. By considering parameters such as radius, length, speed, and flow rate, CFD helps to optimise the different bioprinting parameters to minimise shear stress and decrease cell death during the bioprinting process. Studies have shown that, unlike tapered needles, which require high pressure only at the tip of the nozzle, cylindrical nozzles have a consistent radius and require constant pressure throughout the extrusion, resulting in a longer region of maximum shear stress which, due to the longer transit time in the high shear stress area, leads to increased cell death [31].

Simulation of artificial vessels: Before bioprinting blood vessels, CFD allows the shear stress produced in the walls to be analysed and factors such as vessel diameter, wall thickness, pressure, flow velocity, and viscosity to be predicted and optimised, facilitating the manufacture of functional blood vessels with suitable mechanical and perfusion properties [32].Microfluidic chips: Microfluidic devices, including organs-on-chips, use bioengineering technologies to replicate tissue and organ functions [33]. CFD helps improve precision in micrometre-scale bioprinting by controlling extremely small volumes of fluid [34]. This enables the creation of functional materials and print heads based on microfluidics, allowing for improved precision and geometric control in bioprinting.Vascularisation of tissue fabrication: Vascularisation is crucial in tissue bioprinting to ensure cell survival through adequate perfusion of nutrients and oxygen. CFD plays a key role in analysing flow characteristics such as net force, pressure distribution, shear stress, and oxygen distribution in bioprinted vascular structures [35,36]. This allows for the optimisation of tissue design with perfusable channels prior to bioprinting.

Studies that have used CFD to study bioink in bioprinting have shown that this technique is effective in predicting the behaviour of bioink [7,37]. For example, a study [28] used CFD to study the flow of a hydrogel bioink through a microextrusion nozzle. The results of the study showed that the flow of the bio-ink is affected by the extrusion speed, the nozzle diameter and the viscosity of the bio-ink.

Other work [5] used CFD to study the deposition of a hydrogel bio-ink on a printing surface. The results of the study showed that the deposition of the bio-ink is affected by the extrusion speed, the viscosity of the bio-ink, and the printer surface.

Other studies [4,38] use CFD to study the formation of three-dimensional structures from a hydrogel bio-ink. The results of the study showed that the shape of the three-dimensional structures is affected by the extrusion speed, the viscosity of the bio-ink, and the geometry of the nozzle.

CFD is a promising tool for the optimization of bioimprinting, which can be used to select the optimal bioprinting parameters to obtain three-dimensional structures with the desired properties. It can also be used to predict the behaviour of bioinks under non-ideal bioprinting conditions, such as the presence of cells.

The application of CFD in bioprinting is still at an early stage. However, it has the potential to revolutionise the field of bioprinting but faces a number of important challenges, such as the following:

➢The development of more sophisticated CFD models that take into account the biological effects of cells.➢The validation of CFD models through experimental testing.

CFD has been a valuable tool for testing specific printing parameters, such as nozzle speed, shear stress, printability, and cell viability, allowing the number of repetitions of these tests and the cost and time required to obtain accurate results to be reduced by reducing the number of repetitions required and making the research process more efficient [28].

In spite of this, the number of parameters to be studied, together with their possible combinations, makes it difficult to carry out all the simulations, making it necessary to carry out a design of experiments beforehand. This design of experiments aims to optimise some bioprinting parameters, based on the data collected during this review.

This paper discusses microextrusion bioprinting, pointing out the most recent advances and the state-of-the-art innovations in computational simulation (CFD) for this technology. Through a rigorous methodology and a systematic review, valuable information on CFD in microextrusion bioprinting is compiled. These findings provide crucial information for the design and optimisation of microextrusion systems, driving the development of functional biomimetic devices for biomedical applications.

## 2. Methods

Design of experiments (DoE) is a crucial tool in the optimisation of complex processes that allows for the determination of the optimal conditions and ideal values of the parameters that can affect the extrusion bioprinting process [39]. In order to identify the most favourable conditions during the extrusion bioprinting process, it is necessary to simulate different scenarios to establish the influence of various operational variables in the process and to analyse the possible interactions between them.

### 2.1. Proposal for a Factorial, Central, Composite, Orthogonal, and Rotational Design of Experiments (FCCOR DoE)

It was proposed to use a Factorial, Central, Composite, Orthogonal, and Rotational Design of Experiments (FCCOR DoE) for the optimisation of the bioprinting process. This design combines elements of several DoE methods to maximise the information obtained from a limited number of experiments. Factorial designs allow for the effect of multiple variables to be studied simultaneously [40], while central and composite designs facilitate the estimation of quadratic effects and the construction of response surfaces [41]. Orthogonality ensures that the effects of the variables are estimated independently [42], and routability provides reliable estimates of responses in all directions of the variable space [43].

### 2.2. Determination of the Total Number of Experimental Runs

The total number of experimental runs (N) for an FCCOR DoE design was calculated with the formula N = 2^k^ + 2k + n [44], where *k* is the number of operational variables and n is the number of replications of the central experiment. For example, in the case where *k* was equal to 4 and *n* was 12, to ensure both orthogonality and rotatability of the design, the design should consist of a total of 36 experiments, allowing for the accurate estimation of the effects and interactions between variables.

### 2.3. Axial Distance Definition and Coded Values of Operational Variables

The axial distance must be defined as α = (N_f_)^1/4^ [45], where N_f_ is the number of factorial experiments, i.e., 2^k^. Hence, since k = 4, α was equal to 2, and the coded values of each of the four operational variables of the design were (−2, −1, 0, 1, 2). These values allow the variables to be standardised and facilitate statistical analysis, ensuring that the effects of the variables are measured consistently.

### 2.4. Statistical and Numerical Analysis of Experimental Results

Statistical analysis of the experimental results was performed using Statgraphics Centurion XVI™ software. This analysis included:ANOVA (Analysis of Variance) test: To determine the significance of main effects and interactions [46].Quadratic Regression Analysis: To obtain the response fitting curve and determine the optimal values of the response variables [47].

### 2.5. Graphical Analysis of Experimental Results

The graphical analysis of the experimental results comprises several visual representations that facilitate the interpretation of the data (Figure 1, Figure 2 and Figure 3):Pareto charts: To identify the most significant effects [48].Main Effect Plots: To show the effect of each variable separately [49].Interaction Graphs: To visualise the interactions between variables [50].Response Surface (RS) plots: To show how the response varies as a function of two or more operational variables [51].

These analyses allowed for a comprehensive understanding of the impact of operational variables on the bioprinting process and helped us identify the optimal conditions for process improvement.

## 3. Discussion

### 3.1. Systematic Review Analysis

Following this methodology all papers were obtained in the databases, PubMed, SCOPUS, and WoS, using a series of search strings:“bioprinting AND extrusion AND nozzle”.“bioprinting AND nozzle AND computational”.“bioprinting AND nozzle AND fluid”.“bioprinting AND nozzle AND printhead”.

A total of 74 papers were obtained using these search strings. After that, repeated ones and those that did not match the inclusion criteria were discarded, obtaining a final number of 20 papers to analyse.

From the 20 selected articles, it is shown that since 2015, there has been a considerable increase in research related to computational simulation of bioprinting billets. Figure 4 shows a distribution of the selected articles by country.

Once all papers were analysed, some features could be discussed. After the block division was performed, the first results to discuss were the ones regarding all the features of bioprinting hardware. In this context, from all the papers selected, 14 were original works, and the other 4 papers were general bioprinting reviews. In those reviews [52,53,54,55], general information on bioprinting was provided (Table 1), including an explanation of the technologies types and materials used, among other useful information. In general, most of papers used extrusion bioprinting, with the exception of Carmelo de María et al. [56] and Shi et al. [21] who used inkjet bioprinting or Ponce-Torres et al. [57] who used a gas flow to produce fibres (Table 2).

According to the 14 original papers, there was a huge discrepancy in the geometrical construction of the nozzle. The following three main constructions were used: syringe with a coaxial needle [54,58], syringe with a conical tip [4,21,59], and microfluidics nozzles [60,61]. Other authors used different geometries [56,57,62,63,64,65,66]. In addition, the nozzle diameter varied from 0.2 mm to 1 mm in different works. The studies using a coaxial needle proposed diameters of 810, 710, and 630 μm [54] or 200–450 μm [54], while those ones using a conical tip used 100 μm [21], 500–1000 μm [59] or 440 μm [4]. Studies about microfluidics nozzles had set up different diameters—840, 610, 400, 250, and 200 μm [60] or 1270, 910, and 470 μm [61]. Regarding the nozzle length, Martanto et al. [59] and Reid et al. [67] agreed on using 8.9–10 mm, while Jia Shi et al. [21] and Ponce-Torres et al. [57] used 300 and 900 μm, respectively. Martanto et al. [59] performed a very complete study of geometrical influence in extrusion simulations and one of the most important results in their paper was that they recommended an inner nozzle angle from 20 to 30° to reduce shear stresses. It is important to note that this large range of variation in geometrical measurements makes it difficult for a fair comparison of the results across different studies (Table 2).

Other important factors include the inner coating of the nozzle and the materials used for its construction. This information can be very helpful in computational simulations due to the friction between the bioink and the walls of the nozzle provoking different shear stresses and cell damages. Unfortunately, only Parzel et al. [63] used Ethylene Diamine Tetra-acetic Acid (EDTA) as an inner coating and Shi et al. [21] and Ponce-Torres [57,66] used glass as a construction material. The rest of the authors did not specify any kind of coating or material to be considered.

On the other hand, the most important physical parameters in the bioprinting process are summarized in Table 3 [38,68,69,70]. The data represented are within a range of values obtained from the articles that were selected according to the keywords used in the search (Table 3).

Laboratory atmospheric parameters can have a high impact on the rheological parameters of the used bioink. Among all the analysed papers, the authors presented their results at room temperature [53,63,66] or at 37 °C [54,59,62,67].

The second main block was regarding all the features of computational simulations. In this sense, the nine papers that performed computational simulations of their systems obtained different results that were hardly comparable mainly due to different geometries, materials, boundary conditions, or inlet parameters. Therefore, in this part, only a summarization of the results is going to be presented.

Regarding the boundary conditions, inlet flow was set with volumetric flows values varying from 0.1 to 266 μL/s [4,59,61,67] or a masic flow of 90–100 mg/min [4]. In the same way, Göhl et al. [4] used pressure from 14 to 40 kPa as the inlet pressure of the bioink. Also, some authors showed the viscosity of their bioinks, Shi et al. [21] and Martanto et al. [59] used hydrogels with viscosities varying in the range of 1–9.8 mPa·s. Other authors as Derakshanfar et al. [52], Suntornnond et al. [65] or Kyle et al. [53] used bioinks with higher viscosities on the range 30·10^5^ mPa·s to 30·10^8^ mPa·s (Table 4).

The main results of the CFD simulation works were usually velocities, pressures, and stresses. In this review, velocities varied from 1.6 to 266.6 mm/s [4,54,58,61,62,64,65,66,67] with a single value beyond this range, Shi et al. [21] obtained 3.04 m/s. Pressures varied from 1 to 300 kPa [54,59,62,65,66,67], but several authors concluded that shear stresses must be below 10 kPa [21,53] (Table 4).

Finally, according to the state-of-the-art developments obtained through this review, a design of experiments (DoE) can be proposed. So, this DoE could be used to examine different working parameters of the bioprinting process. As an example, a specific DoE using four operational variables, namely “temperature” (T), “volumetric flow” (V), “pressure” (P), and “viscosity” (v), as presented in the next section, whereas “line width” (W) and “line uniformity” (U) were considered as the response variables.

### 3.2. Optimisation of Bioprinting Parameters Using Taguchi Experimental Design

The success of bioprinting processes depends on the careful selection and optimisation of critical parameters. These parameters, which include temperature (T), volumetric flow rate (V), pressure (P), and viscosity (v), play a key role in determining the quality and reproducibility of bioprinted constructs. To effectively identify the optimal combinations of these parameters, a sound experimental design strategy is essential [71].

Robust parameter design (RPD) is an approach to product realisation activities that focuses on choosing the levels of controllable factors (or parameters) in a process or product to achieve the following two objectives: (1) to ensure that the mean output response is at a desired or target level and (2) to ensure that the variability around this target value is as small as possible. When a DoE study is performed on a process, it is often referred to as a process robustness study [72]. The general DoE problem was developed by a Japanese engineer, Genichi Taguchi, and introduced in the United States in the 1980s [73].

In this study, a Taguchi design approach was used to optimise bioprinting parameters. The Taguchi method is a robust design of experiments technique that allows for the systematic evaluation of multiple parameter combinations with a small number of experiments [74]. This is achieved through the use of an orthogonal matrix, which facilitates the identification of main effects and interactions between factors [75].

To carry out the study, four critical parameters were selected for bioprinting, namely temperature (T), volumetric flow rate (V), pressure (P) and viscosity (v). Each of these parameters was varied at five different levels, taking the previously determined ranges as extremes, as shown below:Temperature (T): 37, 30, 27, 23, 20 °C.Volumetric flow rate (V): 266, 200, 133, 67, 0.1 µL/s.Pressure (P): 40, 33, 27, 20, 14 kPa.Viscosity (v): 30–10^5^, 30–10^6^, 30–10^7^, 30–10^8^, 30–10^9^ mPa·s.

To apply the Taguchi method, the parameter values were coded on a scale from −2 to +2, where each number represents a specific parameter level. The coding is shown below:Level −2: minimum value of the rangeLevel −1: low intermediate valueLevel 0: middle valueLevel +1: intermediate high valueLevel +2: maximum value of the range

An orthogonal matrix L25 (5^4^), suitable for evaluating four factors at five levels each, was used. This matrix consisted of 25 experimental runs, each representing a unique combination of the levels of the four parameters.

For each of the 25 parameter combinations, bioprinting experiments were carried out. The actual parameters for each run were determined from the coded values using the specified scales. Table 1 summarises the maximum, minimum, central, and step values for each parameter.

The results of the experiments were analysed using an analysis of variance (ANOVA) and response analysis to determine the main effects and interactions between parameters, allowing the identification of optimal combinations of parameters that maximised the quality of the bioprinted constructs.

## 4. New Methodologies Applied to Nozzle Design for Different Materials

Nozzles, essential components in a wide range of applications from liquid spraying to 3D printing, have traditionally been designed through trial and error. However, advances in computational simulation, optimisation, and additive manufacturing techniques have opened up a range of new possibilities. The combination of these methodologies is driving the design of more efficient, as accurate and durable nozzles for a wide range of applications.

Thus, the use of different nozzle types is conditioned by the physical parameters of the inks they print, as these condition the printability of the bioinks. Following the flow trend, biomaterial inks are categorised into three viscosity levels—low, medium, and high. The variation in flow with viscosity property appears to be less pronounced in the high viscosity region, which facilitates greater versatility in the printing process [76].

An example was the nozzle designed by Gutiérrez et al., made of 7075T6 aluminium alloy and designed for printing inks based on carbon nanotubes and any nanomaterial such as graphene [77]. This nozzle was designed to integrate a peristaltic pump and solenoid to control the ink flow rate so that it would remain constant.

Another example would be the generation of a medical-grade silicone 3D printing system that uses heat as a curing method. This system consisted of an extruder with a syringe pump to dispense the material, a double-cylinder syringe connected to a static mixer and heatable steel nozzles [78].

These advances not only improve efficiency and precision in printing processes, but also open up new possibilities for innovation in fields as diverse as medicine, electronics, and industrial manufacturing.

## 5. Conclusions

This systematic review has provided a comprehensive overview of the current state of bioprinting nozzle design and use. Key variables affecting nozzle performance have been identified, and common ranges for nozzle diameter and length have been established. In addition, the challenges and opportunities associated with the use of computational fluid dynamics (CFD) simulations to simulate bioprinting processes have been discussed.

It is thus concluded that nozzle design plays a crucial role in bioprinting performance, as it directly influences the quality of the final product. Common ranges for nozzle diameter and length are between 0.2 mm and 1 mm as well as 8 mm and 10 mm, respectively, providing a balance between accuracy and efficiency. Similarly, CFD simulations can be used to predict nozzle performance, although heterogeneity in the configuration of these simulations makes comparisons between different studies difficult. In addition, it is critical that, to minimise cell stress, the shear stress is kept below 10 kPa, which ensures cell viability during the printing process. It has been determined that a better performing nozzle should have an internal angle of 20–30 degrees and an internal EDTA coating, which improves flow and reduces cell damage. Therefore, future research should focus on developing standardised methods for both nozzle design and CFD simulations in order to obtain more consistent and comparable results in the field of bioprinting.

## Figures and Tables

**Figure 1 biomimetics-09-00460-f001:**
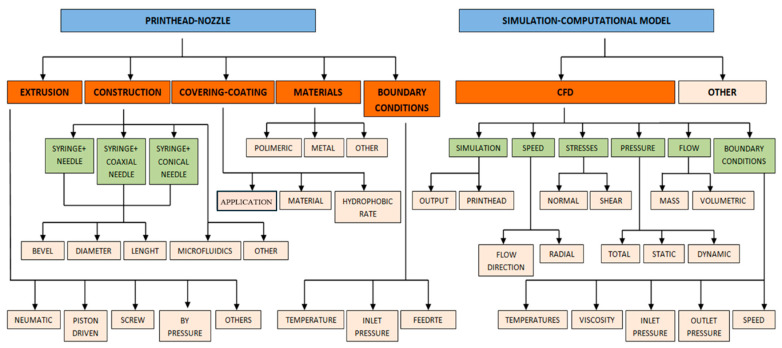
Concept map for data extraction of computational simulation of bioprinting nozzles.

**Figure 2 biomimetics-09-00460-f002:**
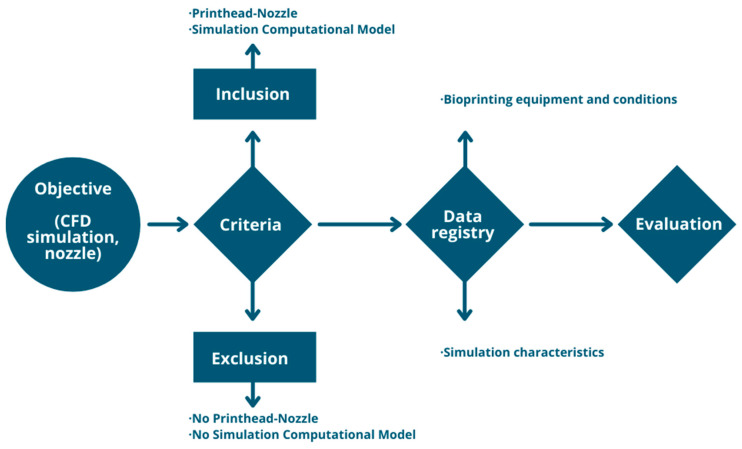
Methodological proposal for bibliographic review.

**Figure 3 biomimetics-09-00460-f003:**
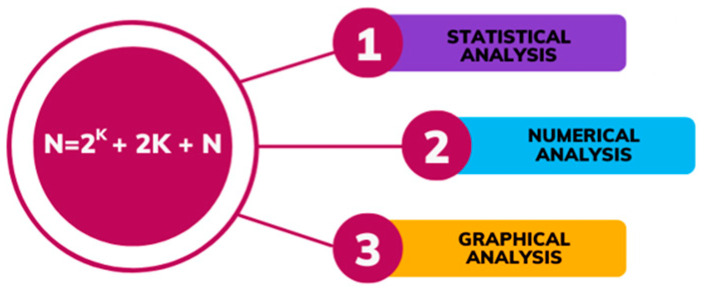
Statistical analysis using Stratgraphics Centurion XVI software; Numerical analysis using ANOVA and quadratic regression analysis; Graphical analysis using Pareto diagram, main effects, interactions and response surfaces.

**Figure 4 biomimetics-09-00460-f004:**
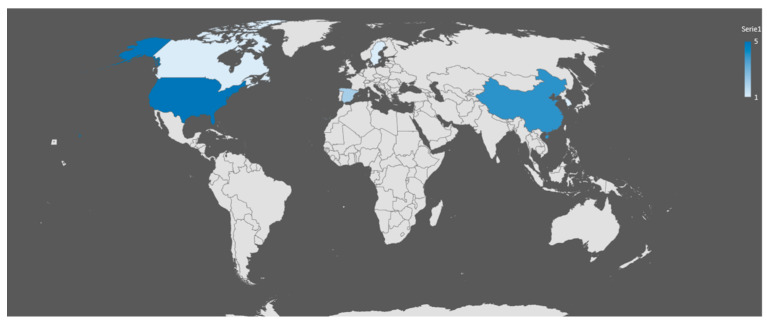
Distribution of articles by country.

**Table 1 biomimetics-09-00460-t001:** Maximum, minimum, central, and step values of the variables.

Variables	Central	Step	Maximum	Minimum
**Temperature (°C)**	28.50	4.25	37	20
**Volumetric flow (μL/s)**	133.050	66.475	266	0.1
**Pressure (kPa)**	27	6.5	40	14
**Viscosity (mPa·s)**	1.515·10^8^	0.7425·10^8^	30·10^8^	30·10^5^

**Table 2 biomimetics-09-00460-t002:** Bioprinting characteristics, nozzle geometry, and computational simulation.

**Characteristics**	Types of bioprinting	Extrusion
		Inkjet
		Gas flow
**Nozzle geometry**	Nozzle types	Coaxial
		Conical
		Microfluidic
		Others
	Nozzle dimensions	Diameter (0.2 mm to 1 mm)
		Length (8.9 mm to 10 mm)
	Nozzle geometry	Internal angle (20° to 30°)
**Computational simulations**		

**Table 3 biomimetics-09-00460-t003:** Physical Parameters of the bioprinting.

**Physical parameters of bioprinting**	Temperature	20 °C to 37 °C
	Volumetric flows	0.1 μL/s to 266 μL/s
	Masic flows	90 mg/min to 100 mg/min
	Pressure	14 kPa to 40 kPa
	Hydrogel viscosity	1 mPa·s to 9.8 mPa·s
	Viscosity	30·105 mPa·s to 30·108 mPa·s

**Table 4 biomimetics-09-00460-t004:** CFD simulation.

**CFD Simulation**	Velocity	1.6 mm/s to 226.6 mm/s
	Pressure	1 kPa to 300 kPa
	Shear stress	<10 kPa

## Data Availability

The authors confirm that the data supporting the findings of this study are available within the article.

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
