# Peer review of "Optimising Bioprinting Nozzles through Computational Modelling and Design of Experiments"

_biomimetics, 2024, doi:10.3390/biomimetics9080460_

Round 1

Reviewer 1 Report (Previous Reviewer 1)

Comments and Suggestions for Authors

The minor revisions that I recommended were adecuately performed except for figues 4 and 8, as they were eliminated. I understand that the removal of these figures was based on a suggestion made to the authors.

Author Response

Reviewer states: The minor revisions that I recommended were adecuately performed except for figues 4 and 8, as they were eliminated. I understand that the removal of these figures was based on a suggestion made to the authors.

We reply: as indicated by the reviewer, figures 4 and 8 were deleted at the suggestion of other reviewers.

Reviewer 2 Report (Previous Reviewer 4)

Comments and Suggestions for Authors

This revised manuscript corrected some serious technical issues from the original manuscript. The readability and quality of this review shows significant improvement. In general, this review provides valuable summary and useful viewpoints to the researchers in this field. However, the format is confusing and looks like a research article instead of a review paper. This issue has been brought up in my previous comments and I believe the authors need to give one more shot to rename the section titles:

"Results" should be replaced with "discussion" or "opinion on optimized nozzle dimension". As this work doesn't generate new data points but simply summarize and identify optimized data from previous studies.

Author Response

Reviewer states: this revised manuscript corrected some serious technical issues from the original manuscript. The readability and quality of this review shows significant improvement. In general, this review provides valuable summary and useful viewpoints to the researchers in this field. However, the format is confusing and looks like a research article instead of a review paper. This issue has been brought up in my previous comments and I believe the authors need to give one more shot to rename the section titles:

"Results" should be replaced with "discussion" or "opinion on optimized nozzle dimension". As this work doesn't generate new data points but simply summarize and identify optimized data from previous studies.

We reply: “3. Results" has been replaced by "3. Discussion", as indicated by the reviewer, in order to correctly express the content included.

Round 2

Reviewer 2 Report (Previous Reviewer 4)

Comments and Suggestions for Authors

The authors have revised the manuscript accordingly. I do not have further comments.

This manuscript is a resubmission of an earlier submission. The following is a list of the peer review reports and author responses from that submission.

Round 1

Reviewer 1 Report

Comments and Suggestions for Authors

The manuscript by Gomez Blanco JC et al presents a systematic review of studies that have utilized computational simulations to optimize nozzle geometry. In bioprinting technology it is well-known that optimizing the bioprinting conditions is crucial to minimize cell damage and improve the viability of printed cells. One of the most important parameters in this regard is the geometry of the nozzles. This work provides an overview of the current state of the art in bioprinting nozzle design, identifying optimal ranges for diameter length, internal angle, and internal coating. Moreover, the authors propose a design of experiments to optimize bioprinting parameters using the data obtained in the review. This work represents a significant advancement in the field of bioprinting and provides researchers with valuable information to further progress in the creation of artificial tissues and organs. For this reason, I recommend accepting the manuscript with the following minor revisions:

In the text, figures must be cited and showed in a sequential order (please cheeck figures 2 and 1)

Figure 2: There is a mistake. When it says “aplication” it should be “application”

Figure 4: Include a title for the graphic and specify the content displayed on the Y-axis.

Figure 8: Include “mm/s” in the velocity range of  1.6-266.6

Table 1: Specify the mesaurements of the parameters showed (temperature, volumetric flow, pressure, and viscosity.

Reviewer 2 Report

Comments and Suggestions for Authors

The publication is a review of current achievements in the field of 3D printing. However, following the trend of printing living counterproducts, the biological part is missing. It would be worth adding a section on how cells respond to variable parameters of the bioprinting process. There are few such studies, and the quality of the manuscript would greatly improve.

Reviewer 3 Report

Comments and Suggestions for Authors

Gomez Blanco J. C. et al, present a bibliometric work that takes into consideration different scientific manuscripts from different search managers regarding the key features of the Bioprinting procedure and the behavior of other parameters. Although the authors present relevant works and a design overview of nozzles in the manuscript, some concerns need to be clarified.

Lines 19: Word EDTA is not described in the abstract

Lines 21 & 22: Perhaps a percentage of the cell damage (minimization) and viability should be included.

Is CFD described correctly? Lines 31 & 32 Continuous Fluid Dynamics is that correct? You mean: Computational Fluid Dynamics

The last paragraph of the introduction does not match the subject of the current journal, so you must rewrite it.

Arrows in “Methodological proposal for bibliographic review.” It seems to be with different lengths or angles. Improve it.

You must follow the order in the figures according to the manuscript. Figure 1 is after Figure 2.

The bioprinting process has evolved over the last five years, and you should include research from 2019 to 2024. There are new methods applied to the design of nozzles and their analysis (CFD) of the flow, pressure, temperature etc.

Your bibliometric analysis should include the country per research that you considered.

I suggest not writing in this way: 0.2-1mm and 8-10mm, which seems to show 0.2 (what?) minus 1 mm; perhaps you should use the following: 0.2 mm to 1 mm. Same in figure 6, figure 7, and figure 8; even which unit is in velocity in Fig. 8?

It is not clear how you applied the Taguchi method; would you mind explain to me in depth? I couldn’t follow when you first mentioned it, in Table 1, line 236.

Line 226: what do you mean by 30x108 and 30x105; they are 3240 and 3150. are you following the SI? Use mathematical notation, same in Table 1 in viscosity values.

Table 1 should include the units per variable.

A discussion section can be placed regarding new methodologies applied to the design of nozzles for different materials. Some physical parameters may be related; here are some references that you may consider discussing:

https://mdpi.com/2306-5354/10/12/1358 - Investigation of Biomaterial Ink Viscosity Properties and Optimization of the Printing Process Based on Pattern Path Planning

https://www.mdpi.com/1996-1944/16/19/6545 - Conceptual Design and Numerical Validation of a Carbon-Based Ink Injector

https://www.mdpi.com/2073-4360/12/5/1031 - 3D Direct Printing of Silicone Meniscus Implant Using a Novel Heat-Cured Extrusion-Based Printer

There are grammar and style errors throughout the manuscript (underlined) that must be rewritten, and the word(s) and sentence(s). 

Finally, a review article requires at least 80 references in order to show the progress of the topic presented comprehensively.

Comments on the Quality of English Language

Extensive editing of the English language is required

Reviewer 4 Report

Comments and Suggestions for Authors

This work uses Continuous Fluid Dynamics (CFD) computation to optimize the nozzle geometry for bioprinting process with minimal cell damage. Although this study aligns with the scope of biomimetics journal, the design of experiment, writing, and research methodologies were too trivial to meet the standard of this journal. Therefore, I recommend rejecting the manuscript and please see my comments below:

1. The introduction did not fully summarize the current stage of bioprinting, only line 29-30 simply mentioned the challenge but did not give enough context. Also, the reference should be more than 1 source to review this field.

2. The major challenge is to develop suitable biomaterials, but this paper is optimizing nozzle geometry, which creates a huge gap to understand the motivation of this study.

3. In the Method section, we assume researchers use proper paper review methods and there is no need to list the details about how you source summarize previous research.

4. Design of experiments can be explained in more details in several sub-sections. Detailed methods about how rheological theories are used and what computational codes/programs/equations were used would be more helpful.

5. Figure 4&5 are not related to the major conclusion of this study.

6. Figure 6-8 can be listed as tables.

7. In conclusion, the authors classify this work as a review. But there are only 22 references...and this manuscript was written in a way as a research article. This is very confusing...